# Clean to Prevent, Monitor to Protect: A Scoping Review on Strategies for Monitoring Cleaning in Hospitals to Prevent HAIs

**DOI:** 10.3390/idr17050120

**Published:** 2025-09-21

**Authors:** Biagio Santella, Antonio Donato, Luigi Fortino, Vittoria Satriani, Rosaria Flora Ferrara, Emanuela Santoro, Walter Longanella, Gianluigi Franci, Mario Capunzo, Giovanni Boccia

**Affiliations:** 1Department of Medicine, Surgery and Dentistry “Scuola Medica Salernitana”, University of Salerno, 84081 Salerno, Italy; bsantella@unisa.it (B.S.); adonato@unisa.it (A.D.); lfortino@unisa.it (L.F.); vsatriani@unisa.it (V.S.); rosferrara@unisa.it (R.F.F.); esantoro@unisa.it (E.S.); gfranci@unisa.it (G.F.); mcapunzo@unisa.it (M.C.); 2A.O.U. San Giovanni di Dio e Ruggi d’Aragona, 84081 Salerno, Italy; wlonganella@unisa.it

**Keywords:** microbiological sampling, healthcare-associated infections, environmental monitoring, hospital hygiene, infection prevention and control

## Abstract

Background/Objectives: Hospital environmental contamination represents a significant source of healthcare-associated infections, yet standardized monitoring approaches are still inconsistent globally. This scoping review aimed to find and assess various tools and strategies used to monitor hospital environmental cleaning and disinfection practices, mapping current evidence and finding research gaps to inform evidence-based recommendations for healthcare facilities. Methods: Following PRISMA Scoping Review guidelines, we conducted comprehensive searches on PubMed and Scopus databases from 2010–2025 using terms related to environmental monitoring, surface sampling, air sampling, and infection control in hospital settings. Eighteen studies met inclusion criteria; data were extracted using standardized forms and synthesized narratively, organizing findings by monitoring approach categories. Results: These studies revealed diverse monitoring approaches including fluorescent markers (22.2%), ATP bioluminescence assays (33.3%), microbiological methods (44.4%), and direct observation techniques (27.8%). MRSA was the most frequently targeted pathogen (55.6%), with limited attention to Gram-negative multidrug-resistant organisms and fungi. Studies showed significant variability in pass/fail thresholds (ATP: 50–500 RLU) and lack of standardized benchmarks. Recent research (50% post-2021) increasingly incorporates molecular techniques and digital technologies, though implementation remains resource intensive. Conclusions: A multimodal approach combining visual inspection, ATP assays, and microbiological methods appears most effective for comprehensive environmental monitoring. Critical gaps include lack of standardized thresholds, limited pathogen diversity focus, and insufficient integration of emerging digital technologies. Future research should focus on setting universal standards, expanding pathogen coverage, and assessing cost-effective monitoring strategies, all while ensuring legal compliance with hygiene regulations to enhance patient safety.

## 1. Introduction

Healthcare-associated infections (HAIs) remain a major challenge in hospitals worldwide [1]. An increasing amount of research suggests that the hospital environment significantly influences the spread and the transmission of pathogenic agents responsible for HAIs. Pathogenic microorganisms, including multidrug-resistant organisms (MDRO), can persist on surfaces and in the air for extended periods, creating reservoirs for potential cross-transmission if not adequately addressed [2]. Microbial contamination can originate from hospitalized patients, family members, and healthcare workers, who, along with airflows, dust and medical devices, represent the most important vectors and vehicles for microbial transfer, otherwise, the risk remains limited [3]. Effective environmental cleaning has been shown to reduce surface contamination and can lead to lower HAI rates [4,5]. Environmental services, encompassing cleaning workers, are essential for infection control. Hospital managers must ensure the implementation and effectiveness of infection control procedures, especially since disinfection services are often outsourced. The use of hospital-grade disinfectants, the careful cleaning of high-touch surfaces, and the routine disinfection of patient rooms and common equipment are essential procedures in mitigating the transmission of HAIs and they are also hygiene requirements for the operation of healthcare activities, especially in high-risk areas such as operating rooms [6,7]. For example, improved cleaning of high-touch surfaces has been associated with decreased transmission of *Clostridioides difficile*, *Staphylococcus aureus*, *Enterococcus* spp. and other pathogens in healthcare facilities. Consequently, routine cleaning and disinfection are now recognized as critical components of infection control programs.

Notwithstanding the existence of international recommendations, including the World Health Organization’s 2019 implementation handbook for preventing and controlling the dissemination of carbapenem-resistant pathogens, hospitals employ various inconsistent methods to monitor and ensure cleaning quality [8]. Numerous institutions continue to depend on conventional visual assessments of cleanliness, which are subjective and frequently unresponsive to microbiological contamination [9]. Healthcare facilities are classified into different risk categories that require customized hygiene approaches. High-risk areas, such as operating rooms, intensive care units, and wards for immunocompromised patients, require more rigorous cleaning protocols and monitoring systems. Conversely, medium- and low-risk areas, such as general wards and administrative spaces, require less intensive but still systematic approaches. This risk-based stratification, established in the seminal work of Dancer (2014) and incorporated into the French, Italian, and German national guidelines, recognizes that microbiological testing is mandatory in high-risk areas to exclude specific pathogens such as *Staphylococcus aureus* and *Aspergillus* spp., while visual and chemical monitoring may be sufficient in low-risk environments [9,10]. Particularly pertinent in practice is the idea that “what gets measured gets done”; crucial cleaning gaps can go undetected in the absence of objective monitoring. Studies using covert observations or quantitative checks have revealed that a large proportion of near-patient surfaces are not cleaned in accordance with hospital policies, even when they appear clean to the eye [11,12]. This gap between perceived and actual cleanliness underscores the need for robust monitoring tools and standardized metrics.

Various tools and strategies have emerged to evaluate hospital environmental hygiene [13,14]. These methods include using fluorescent markers to check how well cleaning is done, ATP bioluminescence tests to see how much organic material is left as a sign of contamination, and microbiological sampling (like culture methods or quick molecular tests) to find specific organisms or overall contaminations and watching cleaning practices directly (either openly or secretly). Each method has its strengths and limitations in terms of sensitivity, practicality, cost, and the type of information provided. However, their comparative effectiveness and optimal use in routine hospital settings are still unclear. There are no universally standard cleanliness guidelines; for instance, different hospitals and studies have applied ATP pass/fail thresholds ranging from 50 to 500 relative light units (RLU) for the same surfaces, and aerobic colony count benchmarks (as <2.5 CFU/cm^2^) are inconsistently used [15]. These inconsistencies make it challenging to compare results across studies or institutions.

Considering these issues, we conducted a comprehensive review of the literature to find and assess the various tools and strategies used to monitor hospital environmental cleaning. We focused on evaluating how each monitoring approach contributes to measuring cleanliness and compliance, with the goal of informing evidence-based recommendations for infection prevention and control teams. The primary aim of this scoping review was to systematically map and synthesize existing evidence on environmental monitoring tools and strategies in hospital settings, guided by the following research questions: (1) What tools and methods are currently used to monitor environmental hygiene in hospitals? (2) What are the main pathogens targeted by environmental monitoring programs? (3) What is the reported effectiveness, advantages, and limitations of different monitoring approaches? (4) What gaps exist in current monitoring practices and what future research directions are needed to improve environmental hygiene surveillance in healthcare settings?

In this study, we summarised many studies to highlight the advantages and disadvantages of each monitoring method and to underscore opportunities for enhancing hospital hygiene practices.

## 2. Materials and Methods

### 2.1. Design

We performed a scoping review in compliance with the PRISMA-ScR guidelines (Preferred Reporting Items for Systematic Reviews and Meta-Analyses extension for Scoping Reviews). This approach was chosen due to the broad range of study designs and outcomes expected on this topic. A review protocol was defined a priori, including the research question, inclusion criteria, and methods for charting and synthesizing data. The protocol was not registered in a public database, which we acknowledge as a limitation of this study.

### 2.2. Search Strategy

A comprehensive literature search was conducted to capture studies that addressed the monitoring or evaluation of hospital cleaning and disinfection practices. The search started in April 2025, covering literature from database inception through 2025, and was conducted on two major databases (PubMed and Scopus). For PubMed, we used the following search string: ((operating room) OR (intensive care unit) OR (high-risk area)) AND ((Microbiological monitoring) OR (Environmental monitoring) OR (Surface sampling) OR (Air sampling)) AND ((Hospital acquired infection) OR (Healthcare associated infection) OR (Infection control)). For Scopus, we applied: ((“operating room” OR “intensive care unit” OR “high-risk area”) AND (“microbiological monitoring” OR “environmental monitoring” OR “surface sampling” OR “air sampling”) AND (“hospital acquired infection” OR “healthcare associated infection” OR “infection control”)). MeSh terms were initially explored, but yielded insufficient results; therefore, free-text keyword combinations were used to ensure comprehensive coverage. The search was supplemented by hand-searching reference lists of relevant papers to find additional studies.

### 2.3. Inclusion and Exclusion Criteria

We included both peer-reviewed research studies and review articles written in English, published between 2010 and 2025, that met the following criteria: (1) focused on healthcare facility surface environments (hospital settings), (2) evaluated tools, methods or strategies for monitoring the cleaning processes, and (3) reported quantitative or qualitative microbial contamination or compliance with cleaning protocols. We included interventional studies (e.g., trials of cleaning interventions with monitoring outcomes), observational studies (e.g., audits or surveys of environmental contamination), and relevant reviews that synthesized data on monitoring methods. Studies solely about hand hygiene, laundry/sterilization of instruments, disinfection methods without a monitoring component (e.g., trials of a disinfectant’s efficacy without assessing monitoring), non-healthcare settings, epidemiology, water, sink and waste, were excluded to keep the scope on monitoring practices. Opinion pieces, letters, and conference abstracts were also excluded unless they contained original data on monitoring tools.

### 2.4. Selection Process

After removing duplicates, two reviewers (B.S. and A.D.) independently screened all titles and abstracts for relevance. Full-text articles were then obtained for all potentially eligible studies and assessed against the inclusion criteria. Disagreements in selection were resolved through discussion. The literature search yielded a total of 3707 records, of which 776 remained after removing duplicates and applying the search strategy (language, date of publication and type of study). After title and abstract screening, 51 articles were identified for full-text review. Among these, 18 studies met all inclusion criteria and were included finally. Figure 1 illustrates the selection process. The included studies were published between 2010 and 2025, reflecting two decades of research into hospital hygiene monitoring.

They encompassed a variety of study designs: eight were interventional or observational studies evaluating monitoring tools in practice (including quasi-experimental studies and prospective surveys/audits), five were systematic or scoping reviews of relevant literature, and the remainder were laboratory or methodological investigations (e.g., comparative assessments of different monitoring techniques).

### 2.5. Data Extraction and Synthesis

A standardized data extraction form was used to chart key information from each included study: authors, year, country, study design, setting (hospital type/unit), the monitoring tool or method evaluated, microorganism target, and main findings. Given the heterogeneity of study designs, a narrative synthesis was undertaken. Given the exploratory nature of this scoping review and the heterogeneous study designs included, formal critical appraisal of individual studies was not performed, consistent with established scoping review methods. To ease comparison, we grouped findings by type of monitoring approach. Specifically, results are organized under categories such as fluorescent marking (FM) for cleaning audits, ATP bioluminescence assays, microbiological methods (culture-based and molecular), direct observation techniques, and other novel or hybrid approaches. Within each category, we summarize the range of findings reported by the studies.

## 3. Results

The 18 selected studies cover a variety of designs, including multicenter trials, observational studies, laboratory investigations, and systematic or scoping reviews [16,17,18,19,20,21,22,23,24,25,26,27,28,29,30,31,32,33]. Critical appraisal of individual studies was not conducted in accordance with scoping review method, which focuses on mapping available evidence rather than assessing study quality. Five studies were cluster-randomized or quasi-experimental trials evaluating interventions for enhanced cleaning and monitoring in intensive care units (ICUs) or operating theatres [16,18,20,28,30,31]. Several studies were prospective observational in design, focusing on environmental contamination mapping [19,21,26,32].

A study compared different ways to check contamination in the air and on surfaces in operating rooms [17], while other studies looked at the overall hospital environment using methods like sequencing, whole-genome sequencing (WGS), or systematic evidence review [22,23,25,27,29,33]. Hospital settings primarily included ICUs [16,19,26,28,30,31,32], operating theatres [17,18], general medical/surgical wards [21,22,24,25], and neonatal ICUs [29]. Geographically, the studies were conducted across diverse countries, including the USA, China, Egypt, Brazil, Cameroon, Italy, and others, reflecting a wide global perspective, though with predominant representation from North America (38.9%) and Europe (22.2%), limited coverage from Asia and South America, and minimal representation from Africa (5.6%). Table 1 summarizes the key information from each included study.

### 3.1. Monitoring Tools and Technologies

Various tools and technologies were used to monitor hospital environmental hygiene. Fluorescent markers (FM) were employed in multiple interventional trials to assess cleaning thoroughness and provide feedback to staff [16,18,20,28]. ATP bioluminescence assays were commonly used as a rapid, quantitative method to assess organic residue on surfaces [20,28,33]. One study compared ATP readings with microbiological cultures and visual inspection, highlighting the moderate concordance among these methods [20].

Many studies commonly used microbiological methods, like counting aerobic colonies from swabs or contact plates [18,21,22,23,25,26,31,32]. Air sampling, both active (via volumetric devices) and passive (settle plates and nitrocellulose membranes), was evaluated to quantify airborne microbial loads in operating theatres and wards [17,21].

In addition, advanced molecular techniques such as 16S rRNA sequencing, whole-genome sequencing (WGS), and qPCR for resistance genes were used to characterize microbial communities and resistance profiles on hospital surfaces [25,26,32]. Finally, disinfection monitoring also looked at new technologies, like pulsed-xenon UV disinfection [30] and antimicrobial polymer coatings on surfaces [22].

### 3.2. Targeted Pathogens in Environmental Contamination

The most studied pathogens were multidrug-resistant organisms (MDRO), including methicillin-resistant *Staphylococcus aureus* (MRSA), *Acinetobacter baumannii*, and *Klebsiella pneumoniae* producing carbapenems (KPC). Several studies specifically targeted MRSA, documenting its presence on high-touch surfaces, in the air, and on equipment in ICUs [19,21,22,29]. One comparative study showed that *A. baumannii* contaminated bed linen more frequently and had a higher transmission potential than MRSA in respiratory ICUs [19]. Sbarra et al. offered sampling guidance for *A. baumannii*, emphasizing its persistence near colonized patients [23]. In Brazil, a multicenter analysis of ICU surfaces revealed a high prevalence of *Staphylococcus* spp., *Streptococcus* spp., *Enterococcus* spp., *Acinetobacter* spp., *Klebsiella* spp., many harboring resistance genes such as bla_KPC_, bla_NDM_, and *mec*A [32]. WGS showed that KPC-producing *Klebsiella pneumoniae* spread in an ICU environment and was closely related to samples taken from patients [26].

### 3.3. Implementation Contexts for Cleaning and Monitoring Interventions

Cleaning interventions and monitoring protocols were deployed in various hospital environments. ICU settings were the most common target due to their high infection risk. In the Ziegler et al. study, interventions were implemented in six ICUs to assess terminal cleaning using ATP and UV markers [28]. The LAMP trial evaluated pulsed-xenon UV after manual cleaning across nine hospitals [30]. In Egypt, enhanced cleaning protocols, including staff training, were introduced in a neurosurgical ICU, resulting in improved surface hygiene and fewer nosocomial infections. In operating theatres, an interventional study used FM and culture-based feedback to improve compliance among cleaning personnel [18], while another assessed air sampling techniques during orthopedic procedures [17]. Several studies addressed surface and air contamination in general wards [21,22,24,25] and neonatal ICUs [29], highlighting unique challenges such as MRSA colonization in mothers and healthcare workers. Monitoring strategies often included staff education, structured feedback, or environmental audits, with emphasis on integrating multimodal approaches for sustainable improvement [16,18,24,28,33].

### 3.4. Reported Outcomes and Effectiveness of Monitoring and Cleaning

Most studies reported that enhanced cleaning protocols and structured monitoring led to improved cleaning performance and reduced environmental contamination. For example, in Carling et al. study across multiple ICUs, the thoroughness of cleaning increased from 49% to 82% after implementing FM-based feedback [16].

Similarly, improved cleaning practices in an operating room (OR) setting led to a reduction in Gram-negative surface contamination from 10.7% to 2.3% [18]. Using an antimicrobial polymer coating lowered the number of surfaces with MRSA from 78% to 11% in a hospital study [22]. In the SHINE trial, ATP-based monitoring significantly decreased MDRO acquisition in ICUs, whereas UV-only feedback showed less impact [28]. In the LAMP trial, using pulsed-xenon UV did not significantly reduce MDRO acquisition compared to the control group, suggesting it might not offer much more benefit than regular manual cleaning [30]. Enhanced cleaning in an Egyptian ICU reduced Gram-negative contamination from 45.6% to 16.3% and decreased nosocomial infections by 39% over six months [31].

A study using WGS showed that even though KPC-producing *K. pneumoniae* often contaminated surfaces, strong infection control measures kept the number of patients who got infected to only two cases [26]. These findings underscore that while environmental reservoirs exist, effective hygiene strategies and containment protocols can prevent cross-transmission.

## 4. Discussion

This scoping review highlights the evolving landscape of environmental hygiene monitoring in hospitals, revealing both significant advances and persistent gaps in our understanding of optimal surveillance strategies. Our analysis of 18 studies spanning 15 years shows that while multiple effective tools exist for monitoring cleaning practices, substantial challenges remain in standardization, global implementation, and integration into comprehensive infection prevention programs. This will help the establishment of universally applicable standards that ensure equitable improvements in hospital cleanliness across all healthcare systems.

The temporal distribution of research in this field reveals interesting patterns that reflect broader trends in infection control. The initial wave of studies (2010–2015, comprising 38.9% of included research) coincided with growing recognition of environmental contamination’s role in healthcare-associated infections [16,17,18]. However, a notable research gap occurred between 2016–2020, with only 11.1% of studies published during this period. The recent surge in research activity (2021–2025, representing 50.0% of studies) likely reflects renewed interest in environmental hygiene following the COVID-19 pandemic, which heightened awareness of environmental transmission pathways and emphasized the critical importance of rigorous cleaning protocols [34,35,36]. Notably, our review indicates a distinct shift in focus among post-2021 studies: compared to earlier years, recent research shows a marked increase in the adoption of molecular techniques—such as sequencing and qPCR—and the first reported integration of digital technologies, supplementing traditional monitoring methods. This trend reflects heightened interest in advanced and automated surveillance approaches following the COVID-19 pandemic.

Fluorescent markers and ATP assays emerged as the most practical, user-friendly options for routine monitoring and frontline feedback, appearing in 22.2% of studies each. From a chemical perspective, fluorescent markers operate through application of photoluminescent compounds that become visible only under ultraviolet light illumination, providing immediate visual confirmation of surface contact during cleaning. ATP bioluminescence assays detect adenosine triphosphate, the universal energy currency present in all living cells and organic residues, through an enzymatic reaction involving luciferase and luciferin that produces quantifiable light emission measured in Relative Light Units (RLU), with higher values indicating greater organic contamination [37,38]. These methods share the crucial advantage of providing immediate or rapid results, which can directly influence cleaning staff behavior in real time [39]. The psychological impact of fluorescent gel auditing cannot be understated—by visually illustrating missed spots under UV light, it transforms abstract cleaning quality concepts into tangible, observable outcomes [40]. Environmental services workers often demonstrate increased engagement when they can see immediate visual validation of their work, creating powerful teachable moments where deficiencies become obvious and corrections can be implemented immediately. Similarly, ATP monitoring transforms cleanliness into concrete numerical values, establishing accountability mechanisms that can be tracked longitudinally to demonstrate improvements or identify problematic areas requiring intervention [41].

Remarkably, our analysis revealed that all five interventional studies incorporating structured feedback mechanisms reported significant improvements in cleaning performance. The multicenter study by Carling et al. conducted across 27 ICUs demonstrated substantial reductions in MDRO transmission, while the cluster-randomized trial by Ziegler et al. showed that ATP feedback effectively reduced MDRO infections and colonizations with only minimal increases in cleaning time [16,28]. This success suggests that the act of measurement itself, combined with systematic feedback, represents a powerful catalyst for behavioral change among cleaning staff.

However, both FM and ATP possess inherent limitations that demand careful consideration during implementation. Neither method provides definitive confirmation of pathogen elimination—a critical distinction often overlooked in routine practice [42]. FM can verify that a surface was physically wiped but cannot guarantee removal of infectious doses of microorganisms. ATP testing, while excellent for detecting organic residue, may be misled when used in isolation since certain high-risk pathogens like *Clostridioides difficile* spores contain minimal ATP yet pose substantial infection risks [43]. The single direct comparison study by Snyder et al. revealed that visual inspection achieved sensitivity rates of only 60–70% compared to these objective measures, highlighting the inadequacy of traditional cleaning assessment methods [20].

Our findings reveal a striking lack of standardization across monitoring practices, representing the most significant barrier to widespread implementation and benchmarking. The variability in what different hospitals or studies consider “clean enough” reflects this challenge—ATP pass/fail thresholds ranging from 50 to 500 RLU for identical surfaces, and inconsistent application of aerobic colony count benchmarks such as <2.5 CFU/cm^2^ [44]. This lack of consensus creates practical difficulties for hospital administrators seeking to implement evidence-based monitoring programs and prevents meaningful comparison of results across institutions or studies. Future research should prioritize the development of international consensus guidelines, potentially through collaborative initiatives involving major infection control organizations such as the Society for Healthcare Epidemiology of America (SHEA), the European Centre for Disease Prevention and Control (ECDC), and the World Health Organization (WHO) [45,46]. Such guidelines should establish evidence-based thresholds that account for surface type, clinical area risk level, and local epidemiological factors.

The cost implications, while rarely quantified in the included studies, represent a crucial consideration for sustainable implementation. Fluorescent markers offer the most economical option for resource-limited settings, requiring minimal initial investment and ongoing costs. ATP monitoring systems demand substantial capital expenditure for equipment purchase and ongoing consumable costs, while microbiological monitoring requires sophisticated laboratory infrastructure and trained personnel. The absence of comprehensive economic evaluations in our reviewed studies represents a significant evidence gap that limits informed decision-making by healthcare administrators, particularly in settings with constrained budgets. The OECD reports clearly assert that investing in preventive measures, especially in environmental hygiene, is a highly effective and cost-efficient strategy. These investments not only safeguard public health, but also yield significant economic benefits over time [47,48].

The microbiological methods employed in 44.4% of studies provide the gold standard for pathogen detection but present practical implementation challenges [49]. Culture-based approaches, while definitive, require 24–48 h for results and specialized laboratory infrastructure [49]. The emerging molecular techniques utilized in recent studies, including 16S rRNA gene sequencing and WGS, offer unprecedented insights into environmental microbiology but remain resource-intensive and require sophisticated technical expertise for implementation and interpretation [32]. These advanced molecular approaches, while promising, face significant barriers to routine implementation including high costs (estimated at $200–500 per sample for comprehensive genomic analysis), need for specialized personnel, and lack of standardized interpretation protocols [50]. However, they offer unique advantages including rapid pathogen identification, antimicrobial resistance profiling, and transmission tracking capabilities that may justify their use in outbreak investigations and high-risk clinical areas. Future cost-effectiveness studies are urgently needed to define optimal use cases for these technologies within routine environmental monitoring programs.

Pathogen specificity emerged as another critical consideration, with MRSA receiving disproportionate attention (55.6% of studies) compared to other clinically relevant organisms [51]. Gram-negative multidrug-resistant organisms, despite their increasing clinical significance, received limited attention, while fungi and viruses—particularly relevant following the COVID-19 pandemic—were entirely absent from environmental monitoring research [52,53]. Future research should adopt a more balanced approach, potentially guided by local epidemiological data and antimicrobial resistance surveillance to ensure monitoring programs address the most relevant pathogen threats in specific healthcare settings. In particular, there is an urgent need to include emerging pathogens such as *Candida auris*, a fungus notable for its high environmental persistence and increasing clinical importance, as well as respiratory viruses beyond SARS-CoV-2. Incorporating these into environmental monitoring programs will better align surveillance efforts with current and evolving infectious risks in healthcare environments.

The geographical distribution of research presents concerning implications for global health equity. High-income countries, particularly the United States (38.9% of studies), dominate the evidence base, while low- and middle-income countries contribute only 27.8% of research despite serving the majority of the World’s population [54]. Africa remains severely underrepresented with only a single study from Cameroon, and large regions of Asia lack any representation. This geographical bias raises critical questions about the generalizability and practical applicability of current evidence in resource-constrained settings where the burden of healthcare-associated infections is often highest.

The integration of monitoring tools into broader infection prevention strategies proved essential for sustained success [13]. Several studies proved that isolated implementation of monitoring tools, even with feedback mechanisms, achieved limited long-term impact without concurrent training programs, clear accountability structures, adequate staffing, and institutional commitment to quality improvement. The most successful interventions combined technical monitoring solutions with comprehensive behavioral change strategies, creating sustainable cultures of environmental hygiene excellence [55].

Digital innovation represents an exciting frontier for environmental monitoring, though current implementation remains limited [56]. Emerging technologies including artificial intelligence-powered image analysis, real-time molecular detection systems, and Internet of Things sensors offer possibilities for continuous, automated monitoring that could revolutionize environmental hygiene surveillance [57,58]. However, these technologies require rigorous evaluation in real-world settings to prove practical effectiveness, cost-efficiency, and user acceptance before widespread adoption. The integration of these digital technologies faces several key challenges including interoperability with existing hospital information systems, data privacy and security concerns, staff training requirements, and initial capital investments. Successful implementation will require phased approaches starting with pilot programs in high-risk areas, followed by gradual system-wide expansion. The development of standardized data formats and communication protocols will be essential to enable seamless integration across different monitoring platforms and healthcare settings.

The ultimate measure of monitoring program success must be improvement in patient outcomes, yet only a minority of studies established clear links between enhanced environmental monitoring and reduced healthcare-associated infection rates [59]. While the biological plausibility of this connection is well-established, the complex multifactorial nature of HAI prevention makes it challenging to isolate the specific contribution of environmental monitoring [60]. Nevertheless, the available evidence consistently supports the role of environmental hygiene in breaking transmission chains, particularly for pathogens with significant environmental persistence [61].

Future research priorities should address the identified evidence gaps through several key initiatives. First, standardization efforts led by international organizations such as WHO or CDC should establish evidence-based thresholds for different monitoring methods across various healthcare settings [46]. Second, implementation research in low- and middle-income countries should evaluate the effectiveness and feasibility of adapted monitoring approaches suitable for resource-constrained environments. Third, comprehensive economic evaluations should quantify the cost-effectiveness of different monitoring strategies to guide resource allocation decisions. Finally, longitudinal studies should examine the sustainability of monitoring programs and their long-term impact on infection rates and patient outcomes.

The evidence strongly supports a paradigm shift toward integrated, multimodal environmental monitoring that combines the immediate feedback capabilities of FM and ATP with the definitive pathogen detection provided by microbiological methods [33]. Successful implementation requires institutional commitment extending beyond technical tool choice to encompass staff training, standardized protocols, regular performance review, and a culture that embraces continuous improvement rather than punitive enforcement. By addressing the identified research gaps and implementing evidence-based monitoring strategies adapted to local contexts and resources, healthcare facilities worldwide can achieve verifiable environmental cleanliness that contributes meaningfully to patient safety and infection prevention.

This review has some limitations, including the restriction to two databases and to English-language publications. However, cross-checking references and the use of major databases help reduce the risk of bias. Additionally, this review has potentially underrepresented outpatient settings, as the focus of included studies was predominantly on operating rooms and other high-risk inpatient areas, which may limit the generalizability of findings to ambulatory care environments.

## 5. Conclusions

Monitoring the effectiveness of hospital cleaning is a critical aspect of infection prevention in healthcare settings. This review shows that in a variety of tools—fluorescent markers, ATP bioluminescence, microbiological cultures, direct observation, and emerging digital technologies—each plays a role in painting a complete picture of environmental hygiene. Fluorescent markers and ATP assays are practical for routine use and staff feedback, microbiological methods provide the definitive assessment of contamination (though mainly for targeted use due to resource demands), and direct observations provide helpful information about compliance and technique. No single method is infallible; thus, a multi-faceted monitoring strategy is recommended. While hand hygiene represents a well-established infection prevention strategy capable of reducing healthcare-associated infections by 50–75%, this critical intervention was excluded from our review scope to maintain focus specifically on environmental surface monitoring. However, it must be acknowledged that hand hygiene monitoring is a parallel and equally critical part of comprehensive infection control programs that call for independent systematic evaluation. Hospitals should strive to incorporate these monitoring tools into a cohesive program, supported by standardized cleaning protocols and ongoing staff education. Doing so creates a cycle of continuous improvement: objective data highlight shortcomings, which can then be addressed through training or process changes, leading to cleaner environments and safer patient care. There is a need for international guidelines or benchmarks to standardize how cleanliness is measured and reported, which would help benchmarking and accelerate adoption of best practices. Additionally, as novel technologies become available, healthcare facilities should evaluate their feasibility and cost-effectiveness, especially in resource-limited contexts, to ensure that advances in monitoring benefit a wide range of health systems globally.

In summary, effective monitoring of hospital environmental hygiene requires balancing thoroughness with practicality. By combining methods and fostering a culture of accountability and excellence in cleaning, hospitals can significantly mitigate one of the key reservoirs for infection transmission.

## Figures and Tables

**Figure 1 idr-17-00120-f001:**
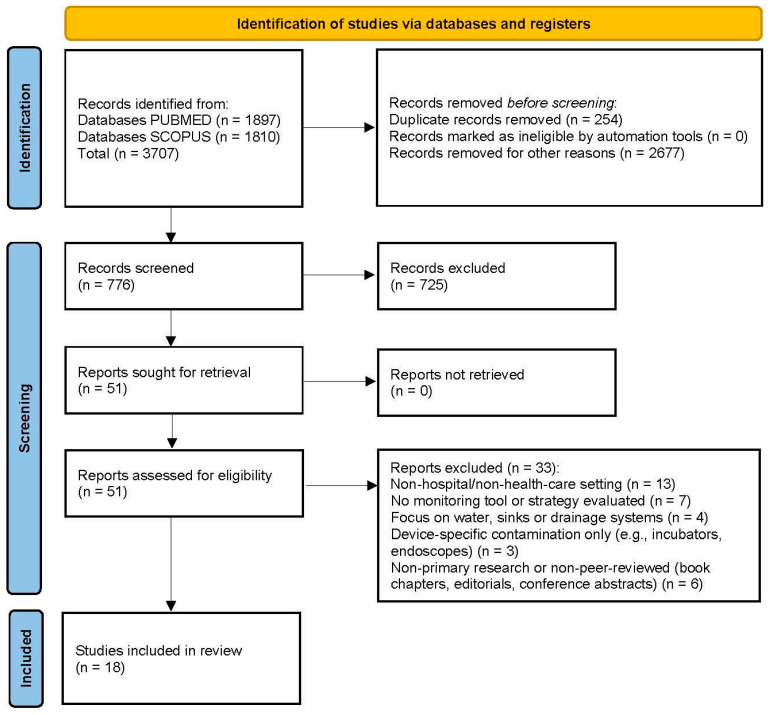
PRISMA 2020 flow diagram illustrating the study selection process.

**Table 1 idr-17-00120-t001:** Summary of the current practices for environmental cleaning in Hospital settings.

Selected References	Setting and Study Design	Monitoring Tools	Pathogens Target	Main Results
Carling et al. (2010) [16]	27 ICUs (USA), multicenter interventional study	Cleaning audit, feedback, infection monitoring	MRSA * VRE MDRO	Enhanced cleaning significantly reduces MDRO transmission
Napoli et al. (2012) [17]	Orthopedics Operating Room (Italy), comparative study	Air sampling: active vs. passive methods	MRSA	Identify differences between air sampling methods; MRSA detected in the air
Munoz-Price et al. (2012) [18]	2 Operating Rooms (USA), pre-post intervention study	Surface microbiological counts, environmental swabs	Bacterial pathogens	Enhanced operation significantly reduces environmental microbial load
Sui et al. (2013) [19]	Inpatient wards (China), prospective observational study	Targeted surface monitoring, environmental swabs	MRSA *A. baumannii*	*A. baumannii* persists longer than MRSA in the environment, with differential risk of transmission
Snyder et al. (2013) [20]	Tertiary care hospital (USA), prospective study	ATP vs. FM vs. Cultures vs. visual inspection	Bacterial pathogens	Visual inspection is not inferior to ATP/FM in predicting cleanliness (sensitivity ~60–70%)
Creamer et al. (2014) [21]	8 acute care wards (Ireland), observational study	Surface swabs, active/passive air plates	MRSA	MRSA is frequently found near patients and in the air; pattern of spread identified
Yuen et al. (2015) [22]	Inpatient ward (Hong Kong), crossover trial	Surface swabs pre/post advanced disinfection	MRSA	Antimicrobial polymer and cleaning routine reduced MRSA compared to standard cleaning (hypochlorite)
Sbarra et al. (2018) [23]	Guidelines (USA), case review	Recommendations for surface sampling	*A. baumannii*	Optimal areas and frequencies for *A. baumannii* sampling suggested
Doll et al. (2018) [24]	Narrative review (USA)	ATP, FM, UV literature summary	*C. difficile* Bacterial pathogens	Multimodal approach recommended; Audit and feedback improve hygiene
Sereia et al. (2021) [25]	University Hospital (Brazil), metagenomics	16S sequencing and surface shotgun metagenomics	*Acinetobacter Enterobacteriaceae* MDRO	Hotspots contaminated by opportunistic pathogens linked to local antibiotic use identified
Wei et al. (2021) [26]	ICUs (China, UK), prospective WGS study	Surface swabs + WGS	KPC	The hospital environment (ICU) contributes to the spread and environmental cloning of KPC strains
Monteiro et al. (2022) [27]	Literature review (Portugal)	Summary of environmental bacterial exposure	MRSA *P. aeruginosa* *C. difficile*	Need for continuous environmental monitoring to reduce microbiological risks highlighted
Ziegler et al. (2022) [28]	6 ICUs (USA), cluster-RCT	ATP vs. UV markers for terminal room cleaning	MRSA VRE *C. difficile* MDRO	ATP feedback effective in reducing MDRO infections/colonizations; slightly increased room cleaning time
Keneh et al. (2024) [29]	Systematic review, NICU (Cameroon)	Summary of environmental MRSA contamination	MRSA	In the neonatal intensive care unit, 16.6% of environmental samples tested positive for MRSA; targeted hygiene measures recommended
Dhar et al. (2024) [30]	9 hospitals (USA), cluster-RCT crossover (LAMP study)	Pulse-xenon UV after terminal cleaning	MRSA VRE MDRO	Pulse UV disinfection does not significantly reduce MDRO safety compared to standard cleaning
Hamed et al. (2024) [31]	Neurosurgery ICU (Egypt), quasi-experimental study	Checklist + enhanced cleaning training; surface swabs	Gram-negative MDRO	Enhanced treatment significantly reduces surface Gram-negative bacteria and decreases the incidence of nosocomial infections
de Bastiani et al. (2024) [32]	41 ICUs (Brazil), multicenter molecular study	16S rRNA sequencing, qPCR for surface resistance genes	Bacterial pathogens resistant	Significant bacterial diversity and the presence of widespread resistance genes identified
Gastaldi et al. (2025) [33]	International scoping review	ATP, FM, microbiological cultures, direct observation, new digital technologies	MRSA VRE *C. difficile* Gram-negative MDRO	Multimodal integration of traditional and innovative techniques for hospital environmental hygiene monitoring recommended

* MRSA “Methicillin-resistant *Staphylococcus aureus*”, VRE “Vancomycin-resistant Enterococci”, MDRO “Multidrug-resistant organism”, KPC “*Klebsiella pneumoniae* carbapenemase”, ICU “Intensive Care Unit”, ATP “adenosine triphosphate”, FM “Fluorescent Markers”, WGS “whole genome sequencing”, Bacterial pathogens “*Staphylococcus aureus*, *Klebsiella pneumoniae*, *Acinetobacter baumannii*, *Pseudomonas aeruginosa*, *Enterobacteriaceae* spp., *Enterococcus* spp., *Clostridioides difficile*”.

## Data Availability

No new data were created in this study.

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
