# Peer review of "Clean to Prevent, Monitor to Protect: A Scoping Review on Strategies for Monitoring Cleaning in Hospitals to Prevent HAIs"

_2036-7449, 2025, doi:10.3390/idr17050120_

Round 1
Reviewer 1 Report
Comments and Suggestions for Authors
The article addresses, as a scoping review, the topic of hospital disinfection and the methods
to evaluate its implementation and effectiveness. The work is of utmost interest, comprehensive, methodologically sound, well supported by bibliographic references, and overall excellent. It offers a useful synthesis in a complex context that, as rightly noted, is scarcely explored. Considering its value, I would like to suggest some refinements to the authors, useful to further strengthen the significance of this work.
Abstract: Cost-effectiveness is perfectly addressed, but I would also suggest including “legal compliance.”
Line 55, regarding “constituting reservoirs,” I would add a sentence describing that vectors (for example, hands) and means (such as airflows, dust, aerosolization) are needed for their transfer; otherwise, the risk remains contained.
Line 58, implementation “and efficacy,” as disinfection services are usually external outsourced contracts to be verified.
Line 62, “and they are also hygiene requirements for the operation of healthcare activities, especially in high-risk areas such as operating rooms.”
In the introduction, to better contextualize the results and discussion, I would insert a brief paragraph describing how there are different studies and hygiene-sanitary requirements (and therefore different scientific production) between high-risk areas and medium- and low-risk areas (see Dancer 2014’s work, which is the basis for French and Italian guidelines). This also serves to highlight how, at least in Italy, France, and Germany, microbiological testing is mandatory in high-risk areas since it is necessary to exclude certain pathogens such as Staphylococcus or Aspergillus.
Search strategy: Is there a risk that the issue related to outpatient settings was not included? The focus seems to be more on operating rooms and high-risk areas. This could be discussed in the text as a limitation.
Inclusion criteria: 1) Focused on the healthcare facility “surface” environment.
Line 223: To be referenced.
Lines 421 and 422: I would suggest a positive expression to emphasize the work, such as “reference cross-check and major database use are methods to reduce the risk of bias.”
Lines 433-434: References to be removed.
Lines 446-447: Sentence not relevant, please remove.
Lines 448-451: Redundant, please remove.
Probably evaluate this reference in addition to Dancer et al 2014, Assadian 2021 https://doi.org/10.1016/j.jhin.2021.03.010
Author Response
|
Comments 1: Abstract: Cost-effectiveness is perfectly addressed, but I would also suggest including “legal compliance.” |
||
|
Response 1: Thanks for pointing this out. We've included it in the abstract. |
||
|
Comments 2: Line 55, regarding “constituting reservoirs,” I would add a sentence describing that vectors (for example, hands) and means (such as airflows, dust, aerosolization) are needed for their transfer; otherwise, the risk remains contained. |
||
|
Response 2: Thanks for your comment. We have incorporated your suggestions and revised the text accordingly. |
||
|
Comments 3: Line 58, implementation “and efficacy,” as disinfection services are usually external outsourced contracts to be verified. |
||
|
Response 3: We agree and have implemented the suggested changes. |
||
|
Comments 4: Line 62, “and they are also hygiene requirements for the operation of healthcare activities, especially in high-risk areas such as operating rooms.” |
||
|
Response 4: Thank you for the sentence, which we have added to the manuscript. |
||
|
Comments 5: In the introduction, to better contextualize the results and discussion, I would insert a brief paragraph describing how there are different studies and hygiene-sanitary requirements (and therefore different scientific production) between high-risk areas and medium- and low-risk areas (see Dancer 2014’s work, which is the basis for French and Italian guidelines). This also serves to highlight how, at least in Italy, France, and Germany, microbiological testing is mandatory in high-risk areas since it is necessary to exclude certain pathogens such as Staphylococcus or Aspergillus. |
||
|
Response 5: Thank you for this suggestion, which allowed us to improve the manuscript's introduction. We have added a detailed description of the sanitation requirements, categorised by different risk areas, considering Dancer (2014) and the suggested study of Assadian et al. (2021) (https://doi.org/10.1016/j.jhin.2021.03.010). We have also added this to the bibliography. |
||
|
Comments 6: Search strategy: Is there a risk that the issue related to outpatient settings was not included? The focus seems to be more on operating rooms and high-risk areas. This could be discussed in the text as a limitation. |
||
|
Response 6: Thank you for your excellent comment. We have expanded the limitations section of the study, incorporating your observations and improving the manuscript. |
||
|
Comments 7: Inclusion criteria: 1) Focused on the healthcare facility “surface” environment. |
||
|
Response 7: Thank you for your insightful observation. We've amended the text as suggested. |
||
|
Comments 8: Line 223: To be referenced. |
||
|
Response 8: Thank you, we have added the reference. |
||
|
Comments 9: Lines 421 and 422: I would suggest a positive expression to emphasize the work, such as “reference cross-check and major database use are methods to reduce the risk of bias.” |
||
|
Response 9: We agree and have refined the sentence as suggested. |
||
|
Comments 10: Lines 433-434: References to be removed. |
||
|
Response 10: Thanks for this point. We have corrected it. |
||
|
Comments 11: Lines 446-447: Sentence not relevant, please remove. |
||
|
Response 11: Thanks for this point. We have removed it. |
||
|
Comments 12: Lines 448-451: Redundant, please remove. |
||
|
Response 12: Thanks for this point. We've removed it. |
||
|
Comments 13: Probably evaluate this reference in addition to Dancer et al 2014, Assadian 2021 https://doi.org/10.1016/j.jhin.2021.03.010 |
||
|
Response 13: Thanks for this point. We have resolved it in Response 5. |

Reviewer 2 Report
Comments and Suggestions for Authors
The review “Clean to Prevent, Monitor to Protect: A Scoping Review on Strategies for Monitoring Cleaning in Hospitals to Prevent HAIs” aims to provide an overview of the strategies used to monitor hospital cleaning and disinfection practices, identifying gaps in research to inform evidence-based recommendations for healthcare facilities.
The strengths of the paper are undoubtedly the abstract and the discussion sections, which are well-structured and comprehensive, while the materials and methods section is decidedly weaker and less clear. Overall, the article has great potential, but some changes should be made to improve its quality and make it suitable for publication in “Infectious Disease Reports”.
- Materials and Methods Section – Search Strategy: This section is not particularly clear. You state that the search was conducted on PubMed and Scopus, combining different keywords, but for PubMed, for example, did you use the MeSH tool, or did you simply combine the keywords manually? The methodology and the combinations used are unclear. A clearer and more detailed version of this section would be desirable.
- Materials and Methods Section – Data Extraction and Synthesis: Which standardized data extraction form was used? Was it taken from previous studies, or is it provided by the PRISMA-ScR guidelines? Please clarify this point.
- In addition to Table 1, it would be useful to include, for example, a graph showing the geographical distribution of the studies included in the scoping review. This would visually demonstrate how certain regions of the world pay more attention to this issue than others. Similarly, a graph highlighting the most commonly used monitoring techniques across studies, or how preferences for specific techniques have changed over time could add further value. The addition of graphs could also help maintain the reader's attention.
- Discussion Section: Fluorescent markers and ATP assays are the most commonly used techniques for routine monitoring in the studies considered. It would be helpful to add a brief explanation of how these tests work from a chemical perspective.
- The studies included in the review conducted on the environmental persistence of Gram-negative bacteria are fewer than those conducted on Gram-positive bacteria. You should highlight more how this gap is actually quite significant, as many HAIs can be caused precisely by Gram-negative pathogens, using recent bibliographic sources.
- References: Please ensure that the reference list follows the guidelines provided by Infectious Disease Reports: https://www.mdpi.com/journal/idr/instructions

Author Response
|
Comments 1: Materials and Methods Section – Search Strategy: This section is not particularly clear. You state that the search was conducted on PubMed and Scopus, combining different keywords, but for PubMed, for example, did you use the MeSH tool, or did you simply combine the keywords manually? The methodology and the combinations used are unclear. A clearer and more detailed version of this section would be desirable. |
||
|
Response 1: Thank you for pointing this out. We agree with this comment. Therefore, we have improved and clarified this section. [For PubMed, we used the following search string: ((operating room) OR (intensive care unit) OR (high-risk area)) AND ((Microbiological monitoring) OR (Environmental monitoring) OR (Surface sampling) OR (Air sampling)) AND ((Hospital acquired infection) OR (Healthcare associated infection) OR (Infection control)). For Scopus, we applied: ( ( "operating room" OR "intensive care unit" OR "high-risk area" ) AND ( "microbiological monitoring" OR "environmental monitoring" OR "surface sampling" OR "air sampling" ) AND ( "hospital acquired infection" OR "healthcare associated infection" OR "infection control" ) ). MeSh terms were initially explored, but yielded insufficient results, therefore free-text keyword combinations were used to ensure comprehensive coverage.] |
||
|
Comments 2: Materials and Methods Section – Data Extraction and Synthesis: Which standardized data extraction form was used? Was it taken from previous studies, or is it provided by the PRISMA-ScR guidelines? Please clarify this point. |
||
|
Response 2: Thank you for this observation. The standardized data extraction form has been adapted to our requests. Additionally, as mentioned in the previous section, we followed the PRISMA-ScR guidelines checklist. Furthermore, we also considered previous studies, including that by Gastaldi et al., which are included in the manuscript, that reported a similar form. |
||
|
Comments 3: In addition to Table 1, it would be useful to include, for example, a graph showing the geographical distribution of the studies included in the scoping review. This would visually demonstrate how certain regions of the world pay more attention to this issue than others. Similarly, a graph highlighting the most commonly used monitoring techniques across studies, or how preferences for specific techniques have changed over time could add further value. The addition of graphs could also help maintain the reader's attention. |
||
|
Response 3: We appreciate the suggestion regarding the visual representation of geographical and temporal distributions. However, we believe that Table 1 already provides comprehensive information on both the geographical origin (country specified for each study) and temporal distribution (year of publication for each study) of the included studies. The geographical diversity is also discussed in the Results section. We think that adding separate figures is unnecessary given the clear presentation of this information in the existing table and text. Moreover, we have added the following text. “[Geographically, the studies were conducted across diverse countries, including the USA, China, Egypt, Brazil, Cameroon, Italy, and others, reflecting a wide global perspective, though with predominant representation from North America (38.9%) and Europe (22.2%), limited coverage from Asia and South America, and minimal representation from Africa (5.6%).]” |
||
|
Comments 4: Discussion Section: Fluorescent markers and ATP assays are the most commonly used techniques for routine monitoring in the studies considered. It would be helpful to add a brief explanation of how these tests work from a chemical perspective. |
||
|
Response 4: We agree and have revised this section to highlight the point you suggested. “[From a chemical perspective, fluorescent markers operate through application of photoluminescent compounds that become visible only under ultraviolet light illumination, providing immediate visual confirmation of surface contact during cleaning. ATP bioluminescence assays detect adenosine triphosphate, the universal energy currency present in all living cells and organic residues, through an enzymatic reaction involving luciferase and luciferin that produces quantifiable light emission measured in Relative Light Units (RLU), with higher values indicating greater organic contamination (PMID: 28900359)]” |
||
|
Comments 5: The studies included in the review conducted on the environmental persistence of Gram-negative bacteria are fewer than those conducted on Gram-positive bacteria. You should highlight more how this gap is actually quite significant, as many HAIs can be caused precisely by Gram-negative pathogens, using recent bibliographic sources. |
||
|
Response 5: Analysis of the pathogens targeted in the included studies reveals a balanced focus on Gram-positive and Gram-negative bacteria. As shown in Table 1, approximately two-thirds of studies included Gram-positive pathogens such as Staphylococcus aureus and Enterococcus spp, while over half targeted Gram-negative bacteria like Acinetobacter baumannii and Klebsiella pneumoniae. Thus, our review reflects an equitable representation of both pathogen groups relevant to hospital environmental contamination and healthcare-associated infections. This supports the comprehensive scope of our review in assessing environmental monitoring strategies across the microbial spectrum. |
||
|
Comments 6: References: Please ensure that the reference list follows the guidelines provided by Infectious Disease Reports: https://www.mdpi.com/journal/idr/instructions |
||
|
Response 6: We appreciate your feedback. We have updated the reference style to comply with the journal's guidelines, as recommended. |

Reviewer 3 Report
Comments and Suggestions for Authors
This work tackles a worldwide concern of public health. The strength of this review paper lies in the importance of the theme – The healthcare-associated infections (HAIs) and the need for effective monitoring of cleaning and disinfection practices.
The authors assessed tools and strategies reported in the literature between 2010 and 2025, to map current evidence and highlight research gaps that could support the establishment of guidelines for healthcare practice.
Indeed, this is a review study with timely relevance.
Please, consider the following comments:
Line 34: “gram-negative” appears with a lowercase “g”. Please correct this to “Gram-negative” (with a capital G), as “Gram” refers to Hans Christian Gram, the author of the Gram staining method.
Line 51,52: The sentence – “An increasing amount of research suggests that the hospital environment significantly influences the transmission of HAIs.” – is more accurate like this: “An increasing amount of research suggests that the hospital environment significantly influences the spread and the transmission of pathogenic agents responsible for HAIs.”
Line 62: Please alter to “Enterococcus spp.”
Line 205 and 343: Please alter to “16S rRNA gene sequencing”
Line 212: The acronym is MDRO not MDROs
Table 1: When is referred “Bacterial pathogens” as bacterial Targets, could the authors be more specific about which bacteria are included? This clarification would be helpful and could be added directly in the table legend.
Line 265: Is the legend of Table1 incomplete or it should be a “.” Instead of “,”?
Line 303: a space is missing – “time[13,16]”
Lines 300-315: In the following sentences, for example, “Carling's multicenter study in 27 ICUs” or “Ziegler's cluster-randomized trial,” the citation format is not consistent with standard scientific writing. Instead, the authors should use formulations such as “In the multicenter study by Carling et al. [XX]…” or “Carling and colleagues’ multicenter study…” to ensure proper attribution and alignment with conventional citation styles.
…and so on, like Ziegler et al or Snyder and colleagues….!
Line 432: “strategy is recommended. (3, 5, 67, 432 70, 28, 61, 71, 67, 12)” – please format references enumeration using [ ].
For clarity, the authors could consider adding a paragraph (where appropriate) to explain that, although hand hygiene is a well-established infection prevention and control strategy capable of reducing up to 50-75% of HAIs when the recommended moments are implemented (there are also equipment and protocols to monitor hand hygiene effectiveness), this topic was not included within the scope of the present study, and why. I understand that this was not the primary focus of the review and that the inclusion criteria are clearly defined at Line 128. However, the importance of hand hygiene should be highlighted and acknowledged, for example, as a parallel subject worthy of independent study and review.
Author Response
|
Comments 1: Line 34: “gram-negative” appears with a lowercase “g”. Please correct this to “Gram-negative” (with a capital G), as “Gram” refers to Hans Christian Gram, the author of the Gram staining method. |
||
|
Response 1: Thanks for pointing that out. We've corrected it. |
||
|
Comments 2: Line 51,52: The sentence – “An increasing amount of research suggests that the hospital environment significantly influences the transmission of HAIs.” – is more accurate like this: “An increasing amount of research suggests that the hospital environment significantly influences the spread and the transmission of pathogenic agents responsible for HAIs.” |
||
|
Response 2: Thank you for your observation. We have made the suggested change. |
||
|
Comments 3: Line 62: Please alter to “Enterococcus spp.” |
||
|
Response 3: Agree. We have done it. |
||
|
Comments 4: Line 205 and 343: Please alter to “16S rRNA gene sequencing” |
||
|
Response 4: Thank you, we have modified it. |
||
|
Comments 5: Line 212: The acronym is MDRO not MDROs |
||
|
Response 4: Thank you, we have modified it. |
||
|
Comments 6: Table 1: When is referred “Bacterial pathogens” as bacterial Targets, could the authors be more specific about which bacteria are included? This clarification would be helpful and could be added directly in the table legend. |
||
|
Response 6: Agree. We have explained it in the table legend. |
||
|
Comments 7: Line 265: Is the legend of Table1 incomplete or it should be a “.” Instead of “,”? |
||
|
Response 7: Agree. We have done it. |
||
|
Comments 8: Line 303: a space is missing – “time[13,16]” |
||
|
Response 8: Thank you. We have done it. |
||
|
Comments 9: Lines 300-315: In the following sentences, for example, “Carling's multicenter study in 27 ICUs” or “Ziegler's cluster-randomized trial,” the citation format is not consistent with standard scientific writing. Instead, the authors should use formulations such as “In the multicenter study by Carling et al. [XX]…” or “Carling and colleagues’ multicenter study…” to ensure proper attribution and alignment with conventional citation styles. …and so on, like Ziegler et al or Snyder and colleagues….! |
||
|
Response 9: Agree. We have changed it. |
||
|
Comments 10: Line 432: “strategy is recommended. (3, 5, 67, 432 70, 28, 61, 71, 67, 12)” – please format references enumeration using [ ]. |
||
|
Response 10: Thank you for this attention. We have removed it. |
||
|
Comments 11: For clarity, the authors could consider adding a paragraph (where appropriate) to explain that, although hand hygiene is a well-established infection prevention and control strategy capable of reducing up to 50-75% of HAIs when the recommended moments are implemented (there are also equipment and protocols to monitor hand hygiene effectiveness), this topic was not included within the scope of the present study, and why. I understand that this was not the primary focus of the review and that the inclusion criteria are clearly defined at Line 128. However, the importance of hand hygiene should be highlighted and acknowledged, for example, as a parallel subject worthy of independent study and review. |
||
|
Response 11: Thanks for your suggestion. We have added a paragraph regarding hand hygiene in the Conclusions. “[While hand hygiene represents a well-established infection prevention strategy capable of reducing healthcare-associated infections by 50-75%, this critical intervention was excluded from our review scope to maintain focus specifically on environmental surface monitoring. However, it must be acknowledged that hand hygiene monitoring constitutes a parallel and equally important component of comprehensive infection control programs that warrants independent systematic evaluation.]” |

Reviewer 4 Report
Comments and Suggestions for Authors
Overall Recommendation: Accept with Major Revisions
- General Summary
The manuscript presents a well-structured and timely scoping review on a topic of considerable importance for infection prevention and control (IPC) in hospital settings. The authors have successfully mapped the current strategies for monitoring environmental cleaning, identifying the most common methods, target pathogens, and critical gaps in the literature, such as the lack of standardization and economic evaluations. The manuscript is well-written, the methodology adheres to the PRISMA-ScR guidelines, and the presentation of results, particularly Table 1, is effective.
However, the review has some important omissions in its literature base and certain areas in the discussion that require further development to maximize its impact. The requested revisions, while significant, are achievable and will substantially improve the quality and completeness of the work.
- Major Comments
2.1. Omission of Relevant Literature
A scoping review must demonstrate a thorough command of the existing literature, including previous reviews. I have identified two pertinent works that have not been cited, and their inclusion could be significant:
- Nante, N., Ceriale, E., Messina, G., Lenzi, D., & Manzi, P. (2017). Effectiveness of ATP bioluminescence to assess hospital cleaning: a review. Journal of preventive medicine and hygiene, 58(3), E177.
- Rationale: This article is a systematic review focused specifically on the use of ATP bioluminescence, one of the central methods discussed in your manuscript (cited as being used in 33.3% of studies). As a review published in 2017, it should be cited in the introduction to contextualize your work and in the discussion to compare your broader findings with their more specific, earlier conclusions. Failing to cite a relevant review on a key sub-topic is a significant omission.
- Fattorini, M., Ceriale, E., Nante, N., Lenzi, D., Manzi, P., Basagni, C., & Messina, G. (2016). Use of a fluorescent marker for assessing hospital bathroom cleanliness. American journal of infection control, 44(9), 1066-1068.
- Rationale: This primary study is an excellent practical example of the use of fluorescent markers, another key method in your review (22.2% of studies). The study not only demonstrates its application but also explores the correlation (or lack thereof) with microbiological methods—a critical point you raise in your discussion. Including this reference would strengthen both the results and discussion sections with a concrete example.
Suggested Revision: to further strengthen the context and discussion, the authors might find it beneficial to integrate the following papers. The review by Nante et al. (2017), being focused on one of the key methods, could be mentioned in the Introduction to more effectively position the unique contribution of this scoping review. Furthermore, both articles offer valuable insights that could enrich the comparative analysis of methods in the Discussion section, thereby increasing the manuscript's depth..
2.2. Discussion of Economic Implications
The authors correctly identify the lack of economic evaluations as a critical gap (lines 330-338). However, the discussion is limited to stating this absence. It could be significantly strengthened by contextualizing the economic importance of combating infections and AMR (Antimicrobial Resistance) by citing high-profile reports. The two OECD reports you mentioned are extremely relevant.
- OECD (2018), Stemming the Superbug Tide: Just a Few Dollars More.
- OECD (2023), Embracing a One Health Framework to Fight Antimicrobial Resistance.
- Rationale: These reports provide robust data on the economic burden of AMR and the cost-effectiveness of prevention interventions, including improved hygiene. Citing them would allow the authors to move from simply stating that "data is lacking" to noting that "while data on the cost-effectiveness of specific monitoring tools is lacking in the primary studies we analyzed, global organizations like the OECD emphasize that investing in hygiene is highly cost-effective, making this research gap even more critical."
Suggested Revision: Integrate these reports into the Discussion section on costs. This will elevate the analysis, providing a macroeconomic context that underscores the urgency of filling the identified research gap.
2.3. Methodological Clarity
In the Search Strategy section (2.2), there are some inconsistencies that undermine the study's reproducibility.
- Search Date: Line 119 states, "The search started in April 2025." This is presumably a typo for "2024".
- Search Period: Line 120 indicates "from database inception through 2025," while line 130 specifies "published between 2010 and 2025." These two statements are conflicting. It is necessary to clarify whether the search was limited to the 2010–2025 period (as suggested by Figure 1 and the subsequent text) or if it began earlier. Consistency is crucial.
Suggested Revision: Correct the start date of the search and standardize the inclusion period for studies throughout the text, ensuring it matches what is reported in Figure 1.
- Minor Comments
3.1. Search Databases
For a scoping review that aims to map the full extent of a field, using only two databases (PubMed and Scopus), while common, may be a limitation.
Suggested Revision: The authors should either justify the choice to limit the search to two databases or, alternatively, mention this as an additional limitation in the study's limitations section. Consideration of databases like Web of Science or CINAHL could have further broadened the search.
3.2. Analysis of Temporal Distribution of Studies
In the Discussion (lines 275-284), the authors note a "research gap" between 2016 and 2020 and a subsequent surge in studies post-2021, which they attribute to the COVID-19 pandemic. This is an astute observation.
Suggested Revision: The analysis could be further enhanced. For instance, do the post-2021 studies show a different focus (e.g., more molecular or digital technologies, a focus on viral pathogens) compared to pre-pandemic ones? A brief reflection on this could add depth to the temporal analysis.
3.3. Specificity of Target Pathogens
The review correctly highlights the disproportionate attention on MRSA (55.6%). The discussion mentions the absence of fungi and viruses (lines 355-358).
Suggested Revision: The authors could more explicitly suggest in the conclusions or future research directions the need for environmental monitoring studies targeting pathogens such as Candida auris—an emerging fungus known for its high environmental persistence—or respiratory viruses beyond SARS-CoV-2.
- Reviewer's Conclusion
This is a high-quality manuscript with great potential. It addresses a relevant problem with a solid methodology. The requested major revisions focus on strengthening the literature base and the contextual depth of the discussion—critical elements for a high-impact review. I am confident that once these points are addressed, the manuscript will represent a significant and highly citable contribution to the field of infection prevention and control.
Therefore, my recommendation is to accept the manuscript following the implementation of major revisions.
Author Response
|
Comments 1: Omission of Relevant Literature: A scoping review must demonstrate a thorough command of the existing literature, including previous reviews. I have identified two pertinent works that have not been cited, and their inclusion could be significant:
Suggested Revision: to further strengthen the context and discussion, the authors might find it beneficial to integrate the following papers. The review by Nante et al. (2017), being focused on one of the key methods, could be mentioned in the Introduction to more effectively position the unique contribution of this scoping review. Furthermore, both articles offer valuable insights that could enrich the comparative analysis of methods in the Discussion section, thereby increasing the manuscript's depth.. |
||
|
Response 1: Thank you for this point. We have strengthened the context and discussion by integrating the papers suggested. |
||
|
Comments 2: Discussion of Economic Implications: The authors correctly identify the lack of economic evaluations as a critical gap (lines 330-338). However, the discussion is limited to stating this absence. It could be significantly strengthened by contextualizing the economic importance of combating infections and AMR (Antimicrobial Resistance) by citing high-profile reports. The two OECD reports you mentioned are extremely relevant.
Suggested Revision: Integrate these reports into the Discussion section on costs. This will elevate the analysis, providing a macroeconomic context that underscores the urgency of filling the identified research gap. |
||
|
Response 2: Agreed. We have integrated the reports into the Discussion section and referenced them as suggested. “[The OECD reports clearly assert that investing in preventive measures, especially in environmental hygiene, is a highly effective and cost-efficient strategy. These investments not only safeguard public health, but also yield significant economic benefits over time.]” |
||
|
Comments 3: Methodological Clarity: In the Search Strategy section (2.2), there are some inconsistencies that undermine the study's reproducibility.
Suggested Revision: Correct the start date of the search and standardize the inclusion period for studies throughout the text, ensuring it matches what is reported in Figure 1. |
||
|
Response 3: Thank you for your comment. We have verified the start date of the search and checked the inclusion period for studies throughout the text. |
||
|
Comments 4: Search Databases: For a scoping review that aims to map the full extent of a field, using only two databases (PubMed and Scopus), while common, may be a limitation. |
||
|
Response 4: Thank you for your insightful comment. We have thoroughly explained our reasoning behind this choice and have included it as a notable limitation in the study limitations section. Your input helps to strengthen our discussion.“[This review has some limitations, including the restriction to two databases and to English-language publications. However, cross-checking references and the use of major databases help reduce the risk of bias.]” |
||
|
Comments 5: Analysis of Temporal Distribution of Studies: In the Discussion (lines 275-284), the authors note a "research gap" between 2016 and 2020 and a subsequent surge in studies post-2021, which they attribute to the COVID-19 pandemic. This is an astute observation. |
||
|
Response 5: Thank you for your great comment. We have enhanced our observation. “[ Notably, our review indicates a distinct shift in focus among post-2021 studies: compared to earlier years, recent research shows a marked increase in the adoption of molecular techniques—sush as sequencing and qPCR—and the first reported integration of digital technologies, supplementing traditional monitoring methods. This trend likely reflects heightened interest in advanced and automated surveillance approaches following the COVID-19 pandemic.]” |
||
|
Comments 6: Specificity of Target Pathogens: The review correctly highlights the disproportionate attention on MRSA (55.6%). The discussion mentions the absence of fungi and viruses (lines 355-358). |
||
|
Response 6: Thank you for your insightful comment. We have integrated into the manuscript, as suggested. ["In particular, there is an urgent need to include emerging pathogens such as Candida auris, a fungus notable for its high environmental persistence and increasing clinical importance, as well as respiratory viruses beyond SARS-CoV-2. Incorporating these into environmental monitoring programs will better align surveillance efforts with current and evolving infectious risks in healthcare environments]”. |

Round 2
Reviewer 2 Report
Comments and Suggestions for Authors
Following the changes made to the text, I believe that the paper is suitable for publication in the journal "Infectious Disease Reports".
Reviewer 4 Report
Comments and Suggestions for Authors
The authors have thoroughly addressed the major concerns raised in the previous review, significantly enhancing the manuscript's depth, contextualization, and methodological clarity. The integration of previously omitted relevant literature, the expanded discussion on economic implications, and the clarification of search strategy inconsistencies have notably improved the quality and completeness of the review.